# Batryticatus Bombyx Protects Dopaminergic Neurons against MPTP-Induced Neurotoxicity by Inhibiting Oxidative Damage

**DOI:** 10.3390/antiox8120574

**Published:** 2019-11-21

**Authors:** Hye-Sun Lim, Joong-Sun Kim, Byeong Cheol Moon, Seung Mok Ryu, Jun Lee, Gunhyuk Park

**Affiliations:** Herbal Medicine Resources Research Center, Korea Institute of Oriental Medicine, 111 Geonjae-ro, Naju-si 58245, Korea; qp1015@kiom.re.kr (H.-S.L.); centraline@kiom.re.kr (J.-S.K.); bcmoon@kiom.re.kr (B.C.M.); smryu@kiom.re.kr (S.M.R.); junlee@kiom.re.kr (J.L.)

**Keywords:** Batryticatus Bombyx, Parkinson’s disease, dopaminergic neuron, neuroprotection

## Abstract

Oxidative stress plays an important role in the degeneration of dopaminergic neurons in Parkinson’s disease (PD). Altered redox homeostasis in neurons interferes with several biological processes, ultimately leading to neuronal death. Oxidative damage has been identified as one of the principal mechanisms underlying the progression of PD. Several studies highlight the key role of superoxide radicals in inducing neuronal toxicity. Batryticatus Bombyx (BB), the dried larva of *Bombyx mori* L. infected by *Beauveria bassiana* (Bals.) Vuill., has been used in traditional medicine for its various pharmacological effects. In the present study, BB showed a beneficial effect on 1-methyl-4-phenyl-1,2,3,6-tetrahydropyridine (MPTP)-induced neurotoxicity by directly targeting dopaminergic neurons. Treatment with BB improved behavioral impairments, protected dopaminergic neurons, and maintained dopamine levels in PD mouse models. Here, we investigated the protective effects of BB on MPTP-induced PD in mice and explored the underlying mechanisms of action, focusing on oxidative signaling. In MPTP-induced PD, BB promoted recovery from impaired movement, prevented dopamine depletion, and protected against dopaminergic neuronal degradation in the substantia nigra pars compacta (SNpc) or the striatum (ST). Moreover, BB upregulated mediators of antioxidative response such as superoxidase dismutase (SOD), catalase (CAT), glutathione (GSH), Heme oxygenase 1 (HO-1), and NAD(P)H (nicotinamide adenine dinucleotide phosphate) dehydrogenase (NQO1). Thus, treatment with BB reduced the oxidative stress, improved behavioral impairments, and protected against dopamine depletion in MPTP-induced toxicity.

## 1. Introduction

Parkinson’s disease (PD) is the second most neurodegenerative disorder induced by a degenerative loss of dopaminergic neurons in projecting from the substantia nigra pars compacta (SNpc) to the striatum (ST), leading to decreased contents of dopamine in the basal ganglia [1]. Approximately 1–2% of the population aged above 65 years and 4% of individuals above 85 years of age are affected by PD [1]. Decreased dopamine contents are associated with adverse movement impairments, including bradykinesia, resting tremor, and postural instability [1,2]. Although the plausible mechanism of PD is unknown, neuronal oxidation, neuroinflammation, and consequent mitochondria-mediated neuronal damage have been implicated in PD pathogenesis [1,3]. Dopaminergic neurons are rich in reactive oxygen species (ROS), which are a major cause of PD due to oxidative stress. ROS is one of the factors that reduces cognitive and motor performance in neurodegenerative diseases. Oxidative stress is associated with a decline in oxidative defense mechanisms mediated by catalase (CAT) and glutathione (GSH), and increased oxidative damage involving hydroxyl radicals and peroxynitrite levels [3,4]. Furthermore, ROS accumulation in PD may be increased by exposure to pesticides and neurotoxins [2,4]. Increased levels of lipid peroxidation products are observed in the SNpc of patients with PD. Thus, a high level of antioxidant activity has been linked to protection against neurodegenerative diseases. Ropinirole (4-[2-(dipropylamino)ethyl]-1,3-dihydro-2H-indol-2-one), a non-ergoline dopamine agonist with chemical structure similar to that of dopamine [5], is used to treat the signs and symptoms of idiopathic PD, such as stiffness, muscle spasms, poor muscle control, and tremors [6]. It is one of the most widely prescribed drugs for PD. We used ropinirole as a positive control to compare the protective effects of dopaminergic neurons associated with Batryticatus Bombyx (BB) against 1-methyl-4-phenyl-1,2,3,6-tetrahydropyridine (MPTP)-induced neurotoxicity.

BB, called Back-Gang-Jam in Korea, is the dried larva of *Bombyx mori* L. infected by *Beauveria bassiana* (Bals.) Vuill. BB, and was originally described in the Chung-bu, category of Dongui-Bogam, an ancient Korean medical book [7]. BB exhibits various pharmacological activities, including anticonvulsant, antiepileptic, and neurotrophic, anticoagulant, antitumor, antibacterial, antifungal, antioxidant, and hypoglycemic effects [7]. Further, BB contains different proteins, peptides, fatty acids, flavonoids, nucleosides, steroids, coumarin, and polysaccharides [7]. Moreover, BB has recently been shown to exhibit neuroprotective and anti-necrotic effects in pyramidal neurons, neurotrophic (upregulation of nerve growth factor) and anti-oxidative (upregulation of superoxide dismutase, SOD) effects in astrocytes, and antiapoptotic effects (inhibition of lipid peroxidation) in hippocampal neurons [7,8,9]. Recently, Hu et al. reported that BB exerts neuroprotective, anti-oxidative, and anti-apoptotic effects against pentylenetetrazole- or H_2_O_2_-induced epileptic neurotoxicity via the PI3K/Akt signaling pathways [10]. However, whether BB has beneficial effects on 1-methyl-4-phenyl-1,2,3,6-tetrahydropyridine (MPTP)-induced neurotoxicity directly by targeting dopaminergic neurons and whether BB improved behavioral impairments, protected dopaminergic neurons, and prevented dopamine depletion in PD mice models has yet to be reported. Further, evidence supporting a mechanistic role of BB in PD is lacking. Thus, we investigated the neuroprotective effects of BB against MPTP-induced neuronal damage and explored the underlying mechanisms.

## 2. Materials and Methods

### 2.1. Preparation of BB Extract

BB was purchased from Kwong-Mung-dang Company (Ulsan, Korea) and authenticated by Dr. Goya Choi (Korea Institute of Oriental Medicine; KIOM, Naju-si, Korea). A voucher specimen (3-18-0030) was deposited at the KIOM. BB was extracted in distilled water for 3 h under reflux (100 ± 2 °C), filtered, evaporated on a rotary vacuum evaporator, and lyophilized (yield, 21.78%). The extracts were stored at −20 °C for further use.

### 2.2. Animals and Drug Administration

Male C57BL/6 mice (8 weeks, 23–24 g, purchased from Doo Yeol Biotech, Seoul, Korea) were maintained at 20–23 °C under 12-h light/12-h dark cycle with food and water provided ad libitum. The experiment was performed according to the guidelines of the Animal Care and Use Committee of KIOM and protocols approved by the Institutional Animal Care Committee of KIOM (KIOM-18-056). Sixty mice were arranged by six groups: (1) Control (*n* = 10); (2) MPTP (*n* = 10); (3) MPTP + BB 1 mg/kg/day (*n* = 10); (4) MPTP + BB 5 mg/kg/day (*n* = 10); (5) MPTP + BB 25 mg/kg/day (*n* = 10); and (6) MPTP + ropinirole 1 mg/kg/day (*n* = 10). BB was administered for 5 consecutive days. MPTP was administered acutely as described previously. On day 3 of the experiment, MPTP (20 mg/kg) was injected intraperitoneally four times at 2-h intervals [3,11]. The control group was administered an equal volume (0.25 mL) of the vehicle.

### 2.3. Rotarod and Pole Test

The rotarod and pole test were performed, as previously described, on days 1 (rotarod) or 5 and 7 (pole) after the last MPTP injection [3,11].

### 2.4. Brain Tissue Preparation

On day 7 after MPTP injection, mice were perfused transcardially with 0.05 M phosphate-buffered saline (PBS). Brains were removed and fixed in 4% PFA (paraformaldehyde) 1 day at 4 °C, and immersed in 30% sucrose for cryoprotection. Serial 30-μm-thick coronal sections were cut on a freezing microtome and stored in 25% glycerol at 4 °C until use. Western blot and kit-based analyses were conducted following rapid dissection, homogenization, and centrifugation of SNpc and ST. The final supernatant was stored at –70 °C until use.

### 2.5. Immunohistochemistry (IHC) Analysis

IHC was performed as described previously [11]. Briefly, treated with 1% H_2_O_2_ (Sigma-Aldrich, St. Louis, MO, USA) for 15 min, and incubated with rabbit anti-tyrosine hydroxylase (TH) (1:1000) overnight at 4 °C. After washing in PBS, the sections were incubated with biotinylated anti-rabbit IgG (Vector Laboratories, Burlingame, CA, USA) (1:200) for 90 min, washed, and incubated with ABC (Vector Laboratories, Burlingame, CA, USA) (1:100) for 1 h. Peroxidase response was visualized with DAB (Sigma-Aldrich, St. Louis, MO, USA). Images were detected using a microscope (Olympus Microscope System BX53; Olympus, Tokyo, Japan) equipped with a 20× objective lens.

### 2.6. Western Blot and Determination of Dopamine Content and SOD, CAT, and GSH Levels

Western blot was performed as described previously [12]. The dopamine content in the ST of the mouse brain was assessed using a commercially available fluorometric assay kit, following the manufacturer’s protocol (Rocky Mountain Diagnostics, Colorado Springs, CO, USA). Further, the levels of SOD, CAT, and GSH expression in the SNpc of the mouse brain were determined using ELISA (enzyme-linked immunosorbent assay) kits, following the manufacturer’s instructions (Biovision, Mountain, CA, USA for SOD and CAT ELISA kits and Cayman chemical, Ann arbor, MI, USA for GSH ELISA kit).

### 2.7. Statistical Analyses

All statistical parameters were calculated using the Graphpad Prism 7.0 software (Graphpad Software, San Diego, CA, USA). Values are expressed as means ± standard error of the mean (S.E.M.). Statistical analyses were performed using one-way analysis of variance (ANOVA) with Tukey’s multiple comparison post-hoc test. *p* < 0.05 was considered statistically significant.

## 3. Results

### 3.1. Effect of BB on MPTP-Induced Behavior Impairment

To evaluate the effect of BB on MPTP-induced motor deficits and postural imbalance, a rotarod test was performed. MPTP significantly reduced the latency time to 16.40 ± 2.99 s on day 1. However, latency times significantly increased in the MPTP + 1–25 mg/kg/day BB and ropinirole groups to 20.60 ± 3.71‒48.61 ± 7.32 s on day 1 (Figure 1). Additionally, to measure the effect of BB on MPTP-induced bradykinesia, a pole test was performed. T-turn (time to turn) and T-LA (locomotion activity time) were significantly increased to 4.62 ± 1.17 s and 10.70 ± 2.19 s, respectively, on day 5. However, T-turn and T-LA were significantly shortened in the MPTP + 25 mg/kg/day BB group to 2.17 ± 0.32 and 6.58 ± 0.61, respectively, on day 5. Further, T-turn and T-LA were significantly increased to 5.56 ± 1.59 s and 11.09 ± 1.94 s, respectively, on day 7, but were significantly shortened in the MPTP + 25 mg/kg/day BB group to 1.44 ± 0.19 and 5.31 ± 0.47, respectively, on day 7 (Figure 1).

### 3.2. Effect of BB on MPTP-Induced Dopamine Depletion

We determined striatal dopamine contents (Figure 2). Injection with MPTP significantly decreased striatal dopamine (to 8.47 ± 0.25 nmol/mL), whereas treatment with 1–25 mg/kg BB increased dopamine contents in ST reduced by MPTP induction (from 13.37 ± 1.57 to 16.79 ± 1.20 nmol/mL).

### 3.3. Effects of BB on MPTP-Induced Dopaminergic Neuronal Loss in SNpc and ST

To confirm the effects of BB on dopaminergic neuronal death, we performed TH-specific IHC in the SNpc and ST of mouse brains. In MPTP-treated mice, the number of TH-positive cells in the SNpc and the optical density (OD) in the ST were decreased to 50.66 ± 3.53% and 43.56 ± 6.16%, respectively, compared with the control group. However, these OD levels were significantly increased by 5–25 mg/kg BB treatment (94.28 ± 13.36% to 95.67 ± 6.69% and 65.65 ± 11.92% to 66.01 ± 2.83%, respectively, compared with the control group) (Figure 3 or Appendix A).

### 3.4. Effects of BB on MPTP-Induced Expression Levels of Antioxidant Enzymes

To evaluate the effects of BB on MPTP-induced expression of antioxidant enzymes, we evaluated SOD, CAT, and GSH levels in the mouse SNpc by ELISA. MPTP significantly decreased SOD (to 43.16 ± 5.74%), CAT (to 2.09 ± 0.17 μM/mL), and GSH (to 0.25 ± 0.04 μM) levels, compared with the control levels, while treatment with 1–25 mg/kg BB increased MPTP-induced decrease in levels of SOD (from 52.05 ± 2.40% to 104.86 ± 8.64%), CAT (from 2.58 ± 0.07 to 2.85 ± 0.11 μM /mL), and GSH (from 0.29 ± 0.09 to 0.89 ± 0.19 μM), compared with the control group (Figure 4). Moreover, MPTP significantly decreased HO-1 (to 40.51 ± 7.12%) and NAD(P)H dehydrogenase (NQO1) (to 45.72 ± 4.29%) levels, compared with the control, while treatment with 1–25 mg/kg BB increased MPTP-induced decrease in levels of HO-1 (from 77.95 ± 8.25% to 98.97 ± 4.54%) and NQO1 (from 63.31 ± 1.02% to 84.62 ± 5.91%), compared with the control group (Figure 5).

## 4. Discussion

We investigated whether BB ameliorated the behavioral impairments and pathology associated with PD using an MPTP-induced PD mouse model. We found that BB attenuated dopamine depletion and inhibited motor impairment in MPTP mice. Moreover, BB increased the number of TH-positive cells in the SNpc and the fiber density in the ST. Further, BB inhibited the downregulation of oxidative stress-related signaling molecules such as SOD, CAT, GSH, heme oxygenase-1 (HO-1), and NAD(P)H dehydrogenase (NQO1) in the SNpc, suggesting a neuroprotective effect.

To evaluate the inhibitory effects of BB on motor impairment, behavioral tests (pole and rotarod tests) used in PD mouse models were conducted in the MPTP mouse model in this study. The rotarod test measures motor coordination, postural balance, and bradykinesia. The pole test determines the agility of animals and measures bradykinesia. Both tests are used to identify behavioral impairments in PD [3,11,13]. Our behavioral analysis revealed that BB significantly improved motor impairment in the MPTP mouse model. Behavior impairments in PD are caused by dopamine deficiency via dopaminergic neurons loss. The most general forms of PD are sporadic with unknown cause, but postmortem studies suggest that apoptosis, oxidative stress, neuroinflammation, and abnormal aggregation are related with dopaminergic neuronal loss [14,15]. In this study, IHC for TH was performed to evaluate the protective effect of BB on dopaminergic neurons in the MPTP model. TH converts tyrosine to dopamine, which is the rate-limiting step in dopamine biosynthesis [16]. Therefore, TH immunoreactivity may accurately reflect the activity of dopaminergic neurons and neurites in animal models of PD [17]. IHC results show that BB protected dopaminergic neurons in both the ST and SNpc, indicating its neuroprotective effect. Moreover, BB significantly restored dopamine levels in the MPTP mouse model. Therefore, BB strongly protects against dopaminergic neuronal degeneration in PD.

Oxidative stress is a central event contributing to the degeneration of dopaminergic neurons in PD pathogenesis [1]. Although ROS production induces PD, the cellular and molecular mechanisms linking oxidative stress to dopaminergic neuronal death are poorly characterized [18,19]. The primary insults induce the greatest spike in ROS levels, which contributes to oxidative damage of lipids, proteins, and nucleic acids, resulting in physiological deficits [19]. Defective macromolecule synthesis or physiological imbalance results in mitochondrial dysfunction and neuroinflammation, which enhances ROS synthesis, resulting in neuronal damage [19]. ROS cytotoxicity is mediated via oxidation of cell constituents, which leads to deterioration of architecture and, finally, death [20]. Under normal conditions, continuous free radical production is neutralized by the protection of antioxidant enzymes, such as SOD, CAT, and GSH [19]. The superoxide radical or singlet oxygen radical generated via cellular metabolism is catalytically converted to H_2_O_2_ and O_2_ by SOD [21]. Toxic H_2_O_2_ in neurons is converted to deleterious hydroxyl radical (OH•) in the presence of Fe^2+^ via Fenton reaction [21,22]. Therefore, CAT reduces H_2_O_2_ to water and molecular oxygen in peroxisomes to curtail free radical-induced neuronal damage [21,23]. However, CAT is absent in mitochondria, where glutathione peroxidase (GPx) catalyzes the reduction of H_2_O_2_ to water and lipid peroxides to their corresponding alcohols [21]. Thus, the antioxidants are involved in first-line defense against oxidative stress [21,22]. Effective first-line defense mediated via SOD, CAT, and GPx is indispensable to the defense strategy, especially the detoxification of superoxide anion radical (O_2_•), which is perpetually generated in normal body metabolism via several processes [21,23]. Oxidative stress may be triggered by reduced efficiency of these endogenous antioxidants, especially in PD patients [21]. Glucose 6-phosphate dehydrogenase is essential for maintaining glutathione levels in the reduced state. A decrease in SOD and other antioxidant enzyme activities and increase in various markers of lipid peroxidation may occur in the SNpc of PD patients [24]. In this study, the activity of SOD, CAT, and GSH was significantly decreased in the group exposed to MPTP only, suggesting that oxidative stress mediates the pathogenesis of PD. BB treatment significantly enhanced the levels of SOD, CAT, and GSH in mice exhibiting MPTP-induced PD.

Further, HO-1 is an enzyme that catalyzes the heme degradation, that generates carbon monoxide, biliverdin, and ferrous iron [25]. HO-1 levels increase by several fold in response to various inter-cellular stresses, including oxidative damage and inflammation, suggesting an important role of this enzyme in cellular protection. Accumulating evidence indicates that HO-1 overexpression protects various tissues from injury, whereas decreased HO-1 expression increases susceptibility to injury due to various stress conditions [25,26]. In the brain, HO-1 expression is low and restricted to small groups of neurons and glial cells [25,27]. Conversely, HO-1 mRNA is physiologically detectable at high levels in the hippocampus and cerebellum, suggesting a cellular reserve of HO-1 transcript that is rapidly available for protein synthesis [28]. It is generally accepted that elevated HO-1 levels restore redox homeostasis and down-modulate inflammation. We previously demonstrated that induction of the enzyme NQO1, which catalyzes the removal of quinone, leads to protection of dopaminergic cells in vitro [29]. Additionally, NQO1 overexpression protected cells against dopamine-induced cell death [30]. The SNpc expresses NQO1 in the brains of normal, as well as PD, patients [31]. Thus, the interaction between various players generates a positive feedback loop, inducing the progressive loss of dopaminergic neurons in PD. Also, understanding the mechanistic system of oxidative stress underlying the dopaminergic neuronal loss may provide a therapeutic approach against PD. Furthermore, in this study, the levels of HO-1 and NQO1 were significantly decreased in the group exposed to MPTP only, suggesting that oxidation was related to the pathogenesis of PD. HO-1 and NQO1 levels significantly improved by BB treatment in MPTP-induced PD mice. Recently, many neuroprotective proteins, including GSH, HO-1, and NQO1, were controlled by Nrf2, a master regulator of antioxidant response in neurons. Studies related to the Nrf2/dopaminergic system reported that Nrf2 activation is the key factor regulating cytoprotective gene expression in knockout Nrf2 mice [32]. Therefore, functional Nrf2 regulation is of major importance in PD-related pathology. In this study, we explored the regulatory effects of BB on MPTP-induced antioxidant signaling by measuring the Nrf2-related proteins HO-1, GSH, and NQO1. The results suggest that BB exerts a therapeutic effect on MPTP-induced oxidative stress, perhaps due to its antioxidant activity and reduced cellular toxicity.

However, the study limitations need to be addressed in future studies. First, studies should focus on in vivo evidence using transgenic animal models to establish the antioxidant potential of BB. Second, we mainly analyzed and separated the components of BB. Studies should isolate the active ingredient by analyzing the components of BB penetrating through the blood–brain barrier, and determine the underlying dopaminergic neuronal protective mechanism. Third, clinical studies are needed to investigate the role of BB and its components in PD prevention/treatment in humans. Based on human equivalent dose calculation, the human dose of the BB extract corresponding to a bodyweight of 60 kg is 121.8 mg/day. Since animal intervention studies differ from randomized clinical trials, it is necessary to adjust and optimize the dosages for animal intervention studies based on clinical trial methodology [33]. Because BB affects multiple signaling pathways of antioxidant response, it may be used in combination therapies for the treatment of PD.

## 5. Conclusions

BB restored impaired movement and dopamine depletion in MPTP-induced PD mice. In addition, BB decreased dopaminergic neuronal loss in the SNpc and ST of MPTP-induced PD mice and increased key indicators of antioxidant response reduced by MPTP. These results strongly suggest that BB is a potential neuroprotective agent that can be used to prevent PD progression.

## Figures and Tables

**Figure 1 antioxidants-08-00574-f001:**
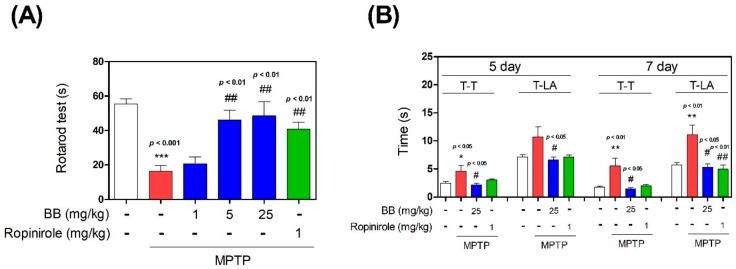
Effects of Batryticatus Bombyx (BB) on 1-methyl-4-phenyl-1,2,3,6-tetrahydropyridine (MPTP)-induced behavior impairment. One day after MPTP treatment, the retention time on the rotating rod was recorded (**A**). On days 5 and 7 after MPTP treatment, time to turn (T-T) completely downward and time to fall off the rod onto the floor (T-LA) were recorded (**B**). Values represent means ± standard error of the mean (S.E.M.). * *p* < 0.05, ** *p* < 0.01, and *** *p* < 0.001 compared with the control group, and # *p* < 0.05 and ## *p* < 0.01 compared with the MPTP-treated group.

**Figure 2 antioxidants-08-00574-f002:**
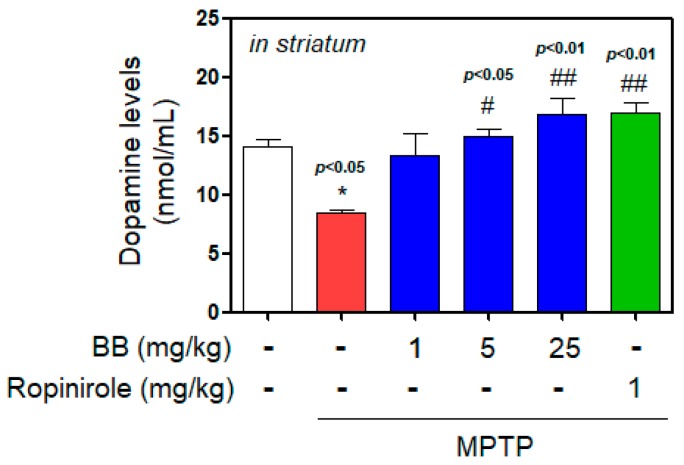
Effects of BB on MPTP-induced dopamine contents. Dopamine contents in the striatum (ST) of mice brain was measured using enzyme-linked immunosorbent assay (ELISA). Values represent means ± S.E.M. * *p* < 0.05 compared with the control group, and # *p* < 0.05 and ## *p* < 0.01 compared with the MPTP-treated group or control group.

**Figure 3 antioxidants-08-00574-f003:**
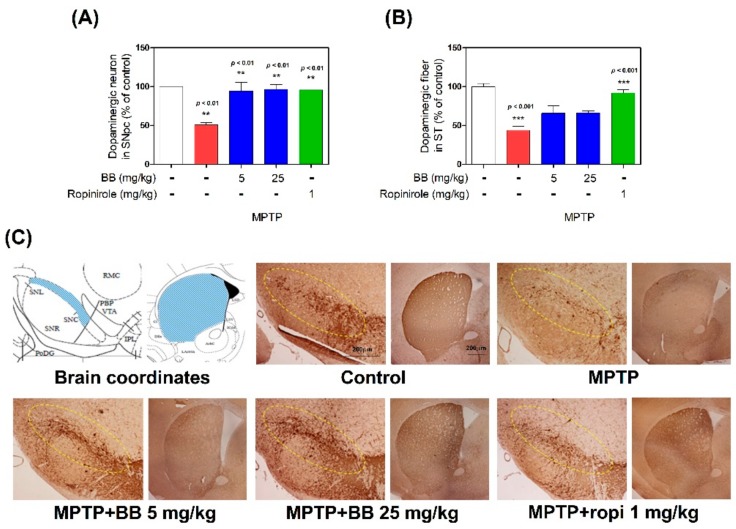
Effects of BB on MPTP-induced dopaminergic neuronal damage. Dopaminergic neurons were visualized by tyrosine hydroxylase (TH)-specific Immunohistochemistry (IHC). The TH-positive neurons in the substantia nigra pars compacta (SNpc) were counted (**A**) and the optical density in the striatum (ST) was measured (**B**). Representative photomicrographs were taken of the SNpc and ST (**C**). Values are presented as means ± S.E.M. ** *p* < 0.01 and *** *p* < 0.001 compared with the control group. Scale bar: 200 µm.

**Figure 4 antioxidants-08-00574-f004:**
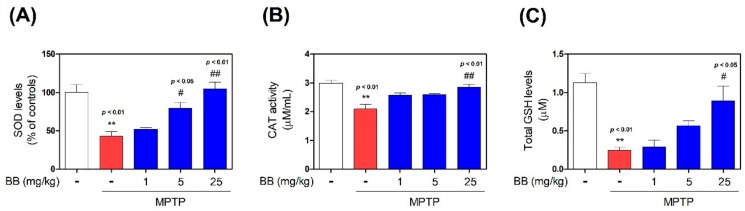
Effects of BB on MPTP-induced antioxidant enzyme levels. Levels of antioxidant enzymes such as superoxidase dismutase (SOD; **A**), catalase (CAT; **B**) and glutathione (GSH; **C**) in SNpc were measured using ELISA kits. Values represent means ± S.E.M. ** *p* < 0.01 compared with the control group, and # *p* < 0.05 and ## *p* < 0.01 compared with the MPTP-treated group.

**Figure 5 antioxidants-08-00574-f005:**
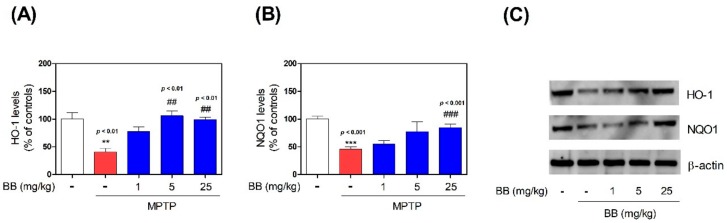
Effects of BB on MPTP-induced Heme oxygenase 1 (HO-1) and NAD(P)H dehydrogenase (NQO1) expression. Bar graphs represent the relative proteins expression of HO-1 (**A**) and NQO1 (**B**) adjusted to β-actin expression. The cellular proteins were used for the detection of HO-1 and NQO1 by Western blot (**C**). Values represent means ± S.E.M. ** *p* < 0.01 and *** *p* < 0.001 compared with the control group, and ## *p* < 0.01 and ### *p* < 0.001 compared with the MPTP-treated group.

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
