# Peer review of "Batryticatus Bombyx Protects Dopaminergic Neurons against MPTP-Induced Neurotoxicity by Inhibiting Oxidative Damage"

_antioxidants, 2019, doi:10.3390/antiox8120574_

Round 1

Reviewer 1 Report

The manuscript is in line with the currently popular trend in neuro-psychiatric medicine to look for the intervention that may diminish the incidence of PD. Data basicly support the conclusions. Experiments have been conducted properly, with appropriate controls, however there are very few issues that I strongly encourage authors to address before the paper is ready to be published

INTRODUCTION: please add a short info on ropinirole

RESULTS

Figure 1. Please clearly indicate in figure A-E parts. I can't see that. Also, what the different colors indicate? Explain controls

Figure 2. as previously, please state what the different colors indicate?Explain controls

Figure 3. as previously, please state what the different colors indicate?Explain controls

Figure 4.as previously, please state what the different colors indicate?Explain controls. Rephrase the title to underline the time when the analyses were done

Figure 5. as previously, please state what the different colors indicate?Explain controls. Adjust WB images to graphs (the bands are not in line with the corresponding bars, which is illegible for analysis)

DISCUSSION: I feel the paragraph on antioxidant response of a cell has a textbook character and in scientific journal shoul be omitted.

Please, add a "limitation" paragraph

At stage of in vivo studies, conclusion should nbe not as "clear".

OVERALL: I strongly encourage to  add info about potential bias in the study following SYRCLE's objectives (https://www.ncbi.nlm.nih.gov/pubmed/24667063)

Author Response

Regarding the reviewer’s comments 1

The manuscript is in line with the currently popular trend in neuro-psychiatric medicine to look for the intervention that may diminish the incidence of PD. Data basicly support the conclusions. Experiments have been conducted properly, with appropriate controls, however there are very few issues that I strongly encourage authors to address before the paper is ready to be published.

INTRODUCTION: please add a short info on ropinirole

à We appreciate your comment. Ropinirole is a pharmacologically active agent used in the treatment of Parkinson's disease (PD) that directly acts on dopamine receptors. It is one of the most widely prescribed drugs for PD. Many researchers have used ropinirole as a positive control. In accordance with the reviewer’s kind comments, we have added an information on ropinirole in the Introduction, as follows:

-Page 2, line 45- 51 in the Introduction section

Ropinirole (4-[2-(dipropylamino)ethyl]-1,3-dihydro-2H-indol-2-one), a non-ergoline dopamine agonist with chemical structure similar to that of dopamine [Coldwell et al., 1999], is used to treat the signs and symptoms of idiopathic PD such as stiffness, muscle spasms, poor muscle control, and tremors [Fukuzaki et al., 2000]. It is one of the most widely prescribed drugs for PD. We used ropinirole as a positive control to compare the protective effects of dopaminergic neurons associated with Batryticatus Bombyx (BB) against MPTP-induced neurotoxicity.

Coldwell MC; Boyfield I; Brown T; Hagan JJ; Middlemiss DN. Comparison of the functional potencies of Ropinirolee and other dopamine receptor agonists at human D2(long), D3 and D4.4 receptors expressed in Chinese hamster ovary cells. British Journal of Pharmacology 1999, 127(7):1696-1702.

Fukuzaki K; Kamenosono T; Nagata R. Effects of Ropinirolee on various parkinsonian models in mice, rats, and cynomolgus monkeys. Pharmacology Biochemistry and Behavior 2000, 65(3):503-508.

RESULTS

Figure 1. Please clearly indicate in figure A-E parts. I can't see that. Also, what the different colors indicate? Explain controls

à We appreciate your comment. We made a mistake. As you might expect, the color of the graph bar is as follows: white for the control group, red for the MPTP group, blue for the MPTP+BB group, and green for the MPTP+ropinirole group. We have corrected the graph as well as added Figure 1 and the figure legend.

-Page 4, line 154-160 in the Results section

Figure 1. Effects of BB on MPTP-induced movement impairment in mice. BB was administered for 5 days. On day 3, at 2 h after BB administration, MPTP was injected four times. One day after MPTP injection, the latency time on the rotating rod was recorded with a-300 s cut-off limit (A). On days 5 and 7 after MPTP injection, time to turn (T-T) completely downward and time to fall off the rod onto the floor (T-LA) were recorded with a 60-s cut-off limit (B). Values represent means ± S.E.M. *p < 0.05, **p < 0.01, and ***p < 0.001 compared with the control group and #p < 0.05 and ## p < 0.01 compared with the MPTP-treated group.

Figure 2. as previously, please state what the different colors indicate? Explain controls. Figure 3. as previously, please state what the different colors indicate? Explain controls.

à We appreciate your comment. We made a mistake. As you might expect, the color of the graph bar is as follows: white for the control group, red for the MPTP group, blue for the MPTP+BB group, and green for the MPTP+ropinirole group. Thank you for your feedback, which have improved our manuscript.

-Page 5, line 166-169 in the Results section

Figure 2. Effects of BB on MPTP-induced dopamine content. Dopamine level in the ST was measured using ELISA. Values represent means ± S.E.M. * p < 0.05 compared with the control group and # p < 0.05 and ## p < 0.01 compared with the MPTP-treated group or control group.

-Page 5, line 177-183 in the Results section

Figure 3. Effects of BB on MPTP-induced dopaminergic neuronal death. Dopaminergic neurons were visualized by TH-specific immunostaining. The number of TH-immunopositive neurons in the SNpc (A) was counted and the optical density in the ST (B) was measured. Representative photomicrographs of the SNpc and ST were taken (C). Values are presented as means ± S.E.M. *p < 0.05 compared with the control group and # p < 0.05 compared with the MPTP-treated group or control group.

Figure 4.as previously, please state what the different colors indicate? Explain controls. Rephrase the title to underline the time when the analyses were done

à We appreciate your comment. We made a mistake. As you might expect, the color of the graph bar is as follows: white for the control group, red for the MPTP group, and blue for the MPTP+BB group. Thank you for your feedback, which have improved our manuscript. In addition, we have corrected the figure and added the figure legend.

-Page 6, line 195-199 in the Results section

Figure 4. Effects of BB on MPTP-induced antioxidant enzyme levels. Levels of antioxidant enzymes such as SOD, CAT, and GSH in SNpc were measured using ELISA kits. Values represent means ± S.E.M. **p < 0.01 compared with the control group and # p < 0.05 and ## p < 0.01 compared with the MPTP-treated group.

Figure 5. as previously, please state what the different colors indicate? Explain controls. Adjust WB images to graphs (the bands are not in line with the corresponding bars, which is illegible for analysis)

à We appreciate your comment. We made a mistake. As you might expect, the color of the graph bar is as follows: white for the control group, red for the MPTP group, and blue for the MPTP+BB group. Thank you for your feedback, which have improved our manuscript. We have corrected Figure 5 and added the legend.

-Page 6, line 200-203 in the Results section

Figure 5. Effects of BB on MPTP-induced HO-1 and NQO1 expression. HO-1 and NQO1 expression in SNpc was measured using western blot. Values represent means ± S.E.M. * p < 0.05 and *** p < 0.001 compared with the control group and # p < 0.05 and ## p < 0.01 compared with the MPTP-treated group.

DISCUSSION: I feel the paragraph on antioxidant response of a cell has a textbook character and in scientific journal shoul be omitted.

Please, add a "limitation" paragraph.

à We appreciate your comment. We have added a paragraph to discuss the limitation in the Discussion section, in accordance with the reviewer's comment.

-Page 8, line 285-296 in the Discussion section

However, the study limitations need to be addressed in future studies. First, studies should focus on in vivo evidence using transgenic animal models to establish the antioxidant potential of BB. Second, we mainly analyzed and separated the components of BB. Studies should isolate the active ingredient by analyzing the components of BB penetrating through the blood-brain barrier, and determine the underlying dopaminergic neuronal protective mechanism. Third, clinical studies are needed to investigate the role of BB and its components in PD prevention/treatment in humans. Based on human equivalent dose calculation, the human dose of the BB extract corresponding to a bodyweight of 60 kg is 121.8 mg/day. Since animal intervention studies differ from randomized clinical trials, it is necessary to adjust and optimize the dosages for animal intervention studies based on clinical trial methodology [33]. Because BB affects multiple signaling pathways of antioxidant response, it may be used in combination therapies for the treatment of PD.

At stage of in vivo studies, conclusion should nbe not as "clear".

à We appreciate your comment. We corrected the Conclusion according to the reviewer's comment.

-Page 8, line 297-301 in the Conclusion section

BB restored impaired movement and dopamine depletion in MPTP-induced PD mice. In addition, BB increased dopaminergic neuronal loss in SNpc and ST of MPTP-induced PD mice and increased key indicators of antioxidant response reduced by MPTP.These results strongly suggest that BB is a potential neuroprotective agent that can be used to prevent PD progression.

OVERALL: I strongly encourage to add info about potential bias in the study following SYRCLE's objectives (https://www.ncbi.nlm.nih.gov/pubmed/24667063)

à We appreciate your comment. We agreed with the reviewer’s comment. We added this content to the Discussion section, referring to PMID:24667063 (Hooijmans CR et al., 2014), in accordance with the reviewer's comment.

- Page 8, line 302-314 in the Discussion section

Based on human equivalent dose calculation, the human dose of the BB extract corresponding to a bodyweight of 60 kg is 121.8 mg/day. Since animal intervention studies differ from randomized clinical trials, it is necessary to adjust and optimize the dosages for animal intervention studies based on clinical trial methodology [Hooijmans CR et al., 2014)]. Because BB affects multiple signaling pathways of antioxidant response, it may be used in combination therapies for the treatment of PD.

Hooijmans CR; Rovers MM; de Vries RB; Leenaars M; Ritskes-Hoitinga M; Langendam MW. SYRCLE's risk of bias tool for animal studies. BMC Medical Research Methodology 2014, 14:43. doi: 10.1186/1471-2288-14-43.

Reviewer 2 Report

Comments to the Authors

In this manuscript, the authors have investigated the protective effects of Batryticatus Bombyx (BB) against 1-methyl-4-phenyl-1,2,3,6-tetrahydropyridine (MPTP)-induced Parkinson’s disease (PD) in mice and provided the underlying mechanism, which is through the activation of antioxidant enzymes. Overall the work is logically organized and the presented data are convincing. However, there are some technical concerns discussed below that the authors should consider to improve the quality of the manuscript.

Regarding the use of BB, how did the authors decide the doses. Did BB at the doses used show any toxicity? Does BB cross BBB to exert the protective effects on dopaminergic neurons?  The drug ropinirole was used along with BB. It would be clearer if the authors explain the purpose of using this drug. Also in some experiments, particularly in behavioral tests, results for MPTP + ropinirole – treated groups are missing. The authors might better consider having this group included in those assays. Figure 1: there are no panel labels and two figure panels were interchanged, which made them not corresponding with the figure legends. The authors described in the legend that MPTP was injected on day 5, 2 h after BB administration but in the materials and methods, it was on day 3. The authors should double-check. Also T-LA time of the MPTP-treated group at day 5 based on the p value does not seem to be statistically different from that of control group as claimed. The authors should better re-run the statistical test for those two groups. Figure 3: The authors should provide better images for TH staining in striata under each condition as they are not clear enough. Also please have p values presented in the graph. In figure 2 and 3, it is very confusing to mention in the figure legends that #p < 0.05 compared with the MPTP-treated group or control group. Does it mean p values were the same when comparing MPTP+BB-treated groups to MPTP-treated group and MPTP+BB-groups vs control? Please clarify this. Figure 5: There are no western blotting results for a loading control. Please have them included. Also the density of NQO-1 bands of MPTP- and MPTP+BB (1 mg/kg)-treated groups appear to be equal (almost undetectable) but in the graph there is a discrepancy. The authors should re-quantify the blots. There is a similar concern in MPTP+BB (5 mg/kg) and MPTP+BB (25 mg/kg) mice. Only at the dose of 25 mg/kg of BB used, the band density of NQO-1 is statistically difference from that of MPTP only group but not at the dose of 5 mg/kg although in the Western blotting result, they look the same.

Minor criticisms:

Names of species including Batryticatus Bombyx, Bombyx mori L., Beauveria bassiana (Bals.) must be in italics. The purpose of the study mentioned in the abstract was to understand the mechanisms underlying the protective effects of BB on MPTP-induced PD in mice with a focus on oxidative signaling. Please briefly explain the rationale for that focus. Line 84, animal and drug administration: please cite the reference which the authors refer to in terms of MPTP regimens. Line 113: The final supernatant was stored at -70oC but not 70oC. Dopamine contents were measured using an ELISA kit. How precise is the method compared to HPLC, the method using widely to measure general neurotransmitters? Please discuss it.

Author Response

Regarding the reviewer’s comments 2

In this manuscript, the authors have investigated the protective effects of Batryticatus Bombyx (BB) against 1-methyl-4-phenyl-1,2,3,6-tetrahydropyridine (MPTP)-induced Parkinson’s disease (PD) in mice and provided the underlying mechanism, which is through the activation of antioxidant enzymes. Overall the work is logically organized and the presented data are convincing. However, there are some technical concerns discussed below that the authors should consider to improve the quality of the manuscript.

Regarding the use of BB, how did the authors decide the doses. Did BB at the doses used show any toxicity?

à We appreciate your comment. In the preliminary study, we have used BB at 25 and 50 mg/kg of as experimental doses. The procedure of the preliminary study was same that of the present study. Dopamine level, which was decreased by MPTP, was increased by BB at both 25 and 50 mf/kg. However, dopamine level was significantly higher in the group treated with 25 mg/kg BB than in the group treated with 50 mg/kg BB. Based on the results of the preliminary studies, we determined 25 mg/kg as the effective dose of BB. BB at this concentration exerted no toxicity. However, because its effect as a single dose has been confirmed, a study of its effect as repeated doses has been planned for safety assessment of BB.

Does BB cross BBB to exert the protective effects on dopaminergic neurons?

à We appreciated your comment. We agreed with the reviewer’s comment. In the present study, BB showed beneficial effect on MPTP-induced neurotoxicity targeting dopaminergic neurons, and BB improved behavioral impairments, protected dopaminergic neurons, and preserved dopamine content in PD mouse models. However, we could not confirm whether BB cross BBB to exert its protective effects on dopaminergic neurons. In a recent study, Xing et al reported that Flavonoids, Amino acid, Lipid, Nucleosides, etc., are a component of BB (Xing et al., 2019). We are using bioactivity-guided fractionation to isolate and analyze the active components of BB that are effective for the Parkinson's disease (Unpublished result). As a follow-up study, we proceed with the analysis of the components of BB, and plan to study the correlation between the protection effect of dopaminergic neurons and the passage of BBB.

Xing D; Shen G; Li Q; Xiao Y; Yang Q; Xia Q. Quality Formation Mechanism of Stiff Silkworm, Bombyx batryticatus Using UPLC-Q-TOF-MS-Based Metabolomics. Molecules 2019, 24(20);E3780. doi: 10.3390/molecules24203780.

The drug ropinirole was used along with BB. It would be clearer if the authors explain the purpose of using this drug. Also in some experiments, particularly in behavioral tests, results for MPTP + ropinirole – treated groups are missing. The authors might better consider having this group included in those assays.

à We appreciate your comment. Ropinirole is a pharmacologically active agent used in the treatment of Parkinson's disease (PD) that directly acts on dopamine receptors. It is one of the most widely prescribed drugs on PD. Many researchers have used ropinirole as a positive control. In accordance with the reviewer’s kind comments, we have added information on ropinirole in the Introduction. In addition, data on behavioral experiments in Figure 1 have been added as follows.

- Page 2, line 45- 51 in the Introduction section

Ropinirole (4-[2-(dipropylamino)ethyl]-1,3-dihydro-2H-indol-2-one), a non-ergoline dopamine agonist with chemical structure similar to that of dopamine [Coldwell et al., 1999], is used to treat the signs and symptoms of idiopathic PD such as stiffness, muscle spasms, poor muscle control, and tremors [Fukuzaki et al., 2000]. It is one of the most widely prescribed drugs for PD. We used ropinirole as a positive control to compare the protective effects of dopaminergic neurons associated with Batryticatus Bombyx (BB) against MPTP-induced neurotoxicity.

Coldwell MC; Boyfield I; Brown T; Hagan JJ; Middlemiss DN. Comparison of the functional potencies of Ropinirolee and other dopamine receptor agonists at human D2(long), D3 and D4.4 receptors expressed in Chinese hamster ovary cells. British Journal of Pharmacology 1999, 127(7):1696-1702.

Fukuzaki K; Kamenosono T; Nagata R. Effects of Ropinirolee on various parkinsonian models in mice, rats, and cynomolgus monkeys. Pharmacology Biochemistry and Behavior 2000, 65(3):503-508.

- Page 4, line 154-160 in the Results section

Figure 1. Effects of BB on MPTP-induced movement impairment in mice. BB was administered for 5 days. On day 3, at 2 h after BB administration, MPTP was injected four times. One day after MPTP injection, the latency time on the rotating rod was recorded with a-300 s cut-off limit (A). On days 5 and 7 after MPTP injection, time to turn (T-T) completely downward and time to fall off the rod onto the floor (T-LA) were recorded with a 60-s cut-off limit (B). Values represent means ± S.E.M. *p < 0.05, **p < 0.01, and ***p < 0.001 compared with the control group and #p < 0.05 and ## p < 0.01 compared with the MPTP-treated group.

Figure 1: there are no panel labels and two figure panels were interchanged, which made them not corresponding with the figure legends. The authors described in the legend that MPTP was injected on day 5, 2 h after BB administration but in the materials and methods, it was on day 3. The authors should double-check. Also T-LA time of the MPTP-treated group at day 5 based on the p value does not seem to be statistically different from that of control group as claimed. The authors should better re-run the statistical test for those two groups.

à We appreciate your comment. Firstly, we made a mistake. We have corrected the figure, and added Figure 1 and the legend. Secondly, we have rechecked the statistical analysis.

- Page 4, line 154-160 in the Results section

Figure 1. Effects of BB on MPTP-induced movement impairment in mice. BB was administered for 5 days. On day 3, at 2 h after BB administration, MPTP was injected four times. One day after MPTP injection, the latency time on the rotating rod was recorded with a-300 s cut-off limit (A). On days 5 and 7 after MPTP injection, time to turn (T-T) completely downward and time to fall off the rod onto the floor (T-LA) were recorded with a 60-s cut-off limit (B). Values represent means ± S.E.M. *p < 0.05, **p < 0.01, and ***p < 0.001 compared with the control group and #p < 0.05 and ## p < 0.01 compared with the MPTP-treated group.

Figure 3: The authors should provide better images for TH staining in striata under each condition as they are not clear enough. Also please have p values presented in the graph. In figure 2 and 3, it is very confusing to mention in the figure legends that #p < 0.05 compared with the MPTP-treated group or control group. Does it mean p values were the same when comparing MPTP+BB-treated groups to MPTP-treated group and MPTP+BB-groups vs control? Please clarify this.

à We appreciate your comment. In accordance with the reviewer’s comment, we have increased the quality of the image of the TH-stained striatum in Figure 3, which we have been enlarged and added to Supplemental Figure 1. We have also corrected Figure 2 and 3.

- Supplemental figure 1

Supplemental figure 1. Effects of BB on MPTP-induced dopaminergic neuronal death. Dopaminergic neurons were visualized by TH-specific immunostaining. Representative photomicrographs of the ST were taken.

-Page 5, line 166-169 in the Results section

Figure 2. Effects of BB on MPTP-induced dopamine content. Dopamine level in the ST was measured using ELISA. Values represent means ± S.E.M. * p < 0.05 compared with the control group and # p < 0.05 and ## p < 0.01 compared with the MPTP-treated group or control group.

-Page 5, line 177-183 in the Results section

Figure 3. Effects of BB on MPTP-induced dopaminergic neuronal death. Dopaminergic neurons were visualized by TH-specific immunostaining. The number of TH-immunopositive neurons in the SNpc (A) was counted and the optical density in the ST (B) was measured. Representative photomicrographs of the SNpc and ST were taken (C). Values are presented as means ± S.E.M. *p < 0.05 compared with the control group and # p < 0.05 compared with the MPTP-treated group or control group.

Figure 5: There are no western blotting results for a loading control. Please have them included. Also the density of NQO-1 bands of MPTP- and MPTP+BB (1 mg/kg)-treated groups appear to be equal (almost undetectable) but in the graph there is a discrepancy. The authors should re-quantify the blots. There is a similar concern in MPTP+BB (5 mg/kg) and MPTP+BB (25 mg/kg) mice. Only at the dose of 25 mg/kg of BB used, the band density of NQO-1 is statistically difference from that of MPTP only group but not at the dose of 5 mg/kg although in the Western blotting result, they look the same.

à We appreciate your comment. We have included the western blotting bands of HO-1 and NQO1, and modified Figure 5 and its legend according to the reviewer's comment.

-Page 6, line 200-203 in the Results section

Figure 5. Effects of BB on MPTP-induced HO-1 and NQO1 expression. HO-1 and NQO1 expression in SNpc was measured using western blot. Values represent means ± S.E.M. * p < 0.05 and *** p < 0.001 compared with the control group and # p < 0.05 and ## p < 0.01 compared with the MPTP-treated group.

Minor criticisms:

Names of species including Batryticatus Bombyx, Bombyx mori L., Beauveria bassiana (Bals.) must be in italics. The purpose of the study mentioned in the abstract was to understand the mechanisms underlying the protective effects of BB on MPTP-induced PD in mice with a focus on oxidative signaling. Please briefly explain the rationale for that focus.

à We appreciate your comment. Batryticatus Bombyx is an herbal name with no italics, and the scientific names "Bombyx mori L" and "Beauveria bassiana" are italicized. In addition, according to the reviewer’s comment, we have revised part of the abstract.

-Page 1, line 13- 22 in the Introduction section

Oxidative damage has been identified as one of the principal mechanisms underlying the progression of PD. Several studies highlight the key role of superoxide radicals in inducing neuronal toxicity. Batryticatus Bombyx (BB), the dried larva of Bombyx mori L. infected by Beauveria bassiana (Bals.) Vuill., has been used in traditional medicine for its various pharmacological effects. In the present study, BB showed a beneficial effect on 1-methyl-4-phenyl-1,2,3,6-tetrahydropyridine (MPTP)-induced neurotoxicity by directly targeting dopaminergic neurons. Treatment with BB improved behavioral impairments, protected dopaminergic neurons, and maintained dopamine levels in PD mouse models. Here, we investigated the protective effects of BB on MPTP-induced PD in mice and explored the underlying mechanisms of action, focusing on oxidative signaling.

Line 84, animal and drug administration: please cite the reference which the authors refer to in terms of MPTP regimens.

à We appreciate your comment. We have indicated a reference to animal and drug administration (Page 2, line 89 in the Materials and Methods section). This changed from line 84 to line 89.

Line 113: The final supernatant was stored at -70oC but not 70oC.

à We corrected it to -70°C according to the reviewer's comment. This changed from line 113 to line 117.

Dopamine contents were measured using an ELISA kit. How precise is the method compared to HPLC, the method using widely to measure general neurotransmitters? Please discuss it.

à Thank you for your very good comment. The dopamine level measured by HPLC in the previous study was 133.33 ± 9.80 ng/mg proteins (control group mouse brain; Choi et al., 2016 and Kim et al., 2013). Although the experiments were conducted in the same way, the current study used an ELISA kit. Based on accurate calculations, the level was 77.40 ± 3.10 ng/mg proteins (control group mouse brain). In conclusion, although ELISA kit is slightly less sensitive than HPLC, we think it has shown very satisfactory results in identifying the trends in dopamine level. In the future, we would like to publish an accurate short communication on this subject with the hope that it will help me in future research.

Choi J; Park G; Kim H; Oh D-S; Kim H; Oh M. In vitro and in vivo neuroprotective effects of walnut (Juglandis semen) in models of Parkinson’s disease. International journal of molecular sciences 2016, 17 (1):108.

Kim HG; Kim TM; Park G; Lee TH; Oh MS. Repeated heat exposure impairs nigrostriatal dopaminergic neurons in mice. Biological and Pharmaceutical Bulletin 2013, 36 (10):1556-1561.

Reviewer 3 Report

This is a descriptive work dealing with the protective effect of Batryticatus Bombyx (BB) against MPTP-induced degeneration/death of dopaminergic neurons. This manuscript is clear and concise, however a few points should be addressed.

Comments

Minor

Line 80. Please define ropinirole. Not all readers are familiar with this compound. 1, legend. There are no A, B, C, D and E panels, only two unlabeled panels. Lines 162, 172, 185, 188, 190, 191 and 192. Please replace "by" with "to". Line 164. Please add the word "decrease" after the words "striatal dopamine". 3C. The light blue structure in the second picture from left should be labeled (presumably striatum). 3, legend. The green bar in A and B should be explained. Lines 187 and 191. Please replace the word "expression" with the words "decrease in levels".

Major

Lines 207-208. Actually, there was no BB-induced increase in the number of TH-positive neurons or fiber density: there was a reduced MPTP-induced decrease in the number of TH-positive neurons, pointing to a protective effect of BB towards existing neurons. Lines 265-267. "Increased HO-1 levels constitute an anatomopathological feature of many neurological diseases, such as neurodegenerative disorders and brain infections, which correlate with exacerbated oxidative stress and inflammation [23,24]. ". However, Fig. 5A shows that MPTP treatment causes a decrease in the expression level of HO-1. The authors should explain this discrepancy.

Author Response

Regarding the reviewer’s comments 3

This is a descriptive work dealing with the protective effect of Batryticatus Bombyx (BB) against MPTP-induced degeneration/death of dopaminergic neurons. This manuscript is clear and concise, however a few points should be addressed.

Minor

Line 80. Please define ropinirole. Not all readers are familiar with this compound.

à We appreciate your comment. Ropinirole is a pharmacologically active agent used in the treatment of Parkinson's disease (PD) that directly acts on dopamine receptors. It is one of the most widely prescribed drugs on PD. Many researchers have used ropinirole as a positive control. In accordance with the reviewer’s kind comments, we have added information on ropinirole in the Introduction as follows.

- Page 2, line 45- 51 in the Introduction section

Ropinirole (4-[2-(dipropylamino)ethyl]-1,3-dihydro-2H-indol-2-one), a non-ergoline dopamine agonist with chemical structure similar to that of dopamine [Coldwell et al., 1999], is used to treat the signs and symptoms of idiopathic PD such as stiffness, muscle spasms, poor muscle control, and tremors [Fukuzaki et al., 2000]. It is one of the most widely prescribed drugs for PD. We used ropinirole as a positive control to compare the protective effects of dopaminergic neurons associated with Batryticatus Bombyx (BB) against MPTP-induced neurotoxicity.

Coldwell MC; Boyfield I; Brown T; Hagan JJ; Middlemiss DN. Comparison of the functional potencies of Ropinirolee and other dopamine receptor agonists at human D2(long), D3 and D4.4 receptors expressed in Chinese hamster ovary cells. British Journal of Pharmacology 1999, 127(7):1696-1702.

Fukuzaki K; Kamenosono T; Nagata R. Effects of Ropinirolee on various parkinsonian models in mice, rats, and cynomolgus monkeys. Pharmacology Biochemistry and Behavior 2000, 65(3):503-508.

1, legend. There are no A, B, C, D and E panels, only two unlabeled panels.

à We appreciate your comment. Firstly, we made a mistake. We have corrected the graphs and added Figure 1 and its legend. Secondly, we have rechecked the statistical analyses.

- Page 4, line 154-160 in the Results section

Figure 1. Effects of BB on MPTP-induced movement impairment in mice. BB was administered for 5 days. On day 3, at 2 h after BB administration, MPTP was injected four times. One day after MPTP injection, the latency time on the rotating rod was recorded with a-300 s cut-off limit (A). On days 5 and 7 after MPTP injection, time to turn (T-T) completely downward and time to fall off the rod onto the floor (T-LA) were recorded with a 60-s cut-off limit (B). Values represent means ± S.E.M. *p < 0.05, **p < 0.01, and ***p < 0.001 compared with the control group and #p < 0.05 and ## p < 0.01 compared with the MPTP-treated group.

Lines 162, 172, 185, 188, 190, 191 and 192. Please replace "by" with "to".

à We modified "by" on line 163, 173, 186, 187, 191 to "to" and "by" on line 165, 188, 189, 192, 193 to "from".

Line 164. Please add the word "decrease" after the words "striatal dopamine".

à We added "decrease" to line 165. This changed from line 164 to line 165.

3C. The light blue structure in the second picture from left should be labeled (presumably striatum). 3, legend. The green bar in A and B should be explained.

à According to the reviewer’s kind comments, we modified Figure 3.

-Page 5, line 181-187 in the Results section

Figure 3. Effects of BB on MPTP-induced dopaminergic neuronal death. Dopaminergic neurons were visualized by TH-specific immunostaining. The number of TH-immunopositive neurons in the SNpc (A) was counted and the optical density in the ST (B) was measured. Representative photomicrographs of the SNpc and ST were taken (C). Values are presented as means ± S.E.M. *p < 0.05 compared with the control group and # p < 0.05 compared with the MPTP-treated group or control group.

Lines 187 and 191. Please replace the word "expression" with the words "decrease in levels".

à We modified “expression” to “decrease in levels”. This changed from line 187 and 191 to line 188 and 192.

Major

Lines 207-208. Actually, there was no BB-induced increase in the number of TH-positive neurons or fiber density: there was a reduced MPTP-induced decrease in the number of TH-positive neurons, pointing to a protective effect of BB towards existing neurons. Lines 265-267. "Increased HO-1 levels constitute an anatomopathological feature of many neurological diseases, such as neurodegenerative disorders and brain infections, which correlate with exacerbated oxidative stress and inflammation [23,24]. ". However, Fig. 5A shows that MPTP treatment causes a decrease in the expression level of HO-1. The authors should explain this discrepancy.

à We are very grateful for your comment. We made a mistake in writing the sentence. We modified the sentence as follows.

-Page 7, line 258-262 in the Discussion section

HO-1 expression is increased several fold in response to a variety of cellular stresses and stimuli, including oxidative stress and inflammation, suggesting an important role of this enzyme in tissue protection. Accumulating evidence indicates that HO-1 overexpression protects against injury in tissues, while reduced HO-1 levels increase susceptibility to injury under stress conditions [25,26].

Round 2

Reviewer 1 Report

The authors followed majority of my comments and I think the article is ready to be published.

However, you could report your efforts to avoid selection bias, performance bias, attrition bias, detection bias and reporting bias - these are isues I was asking for during the first review.

Author Response

Regarding the reviewer’s comments 1

The authors followed majority of my comments and I think the article is ready to be published.

However, you could report your efforts to avoid selection bias, performance bias, attrition bias, detection bias and reporting bias - these are isues I was asking for during the first review.

à We appreciate your comment. We agreed with the reviewer’s comment. Prejudice against a person or thing prevents us from acquiring objective and accurate information about it. A prejudiced person refuses to accept the contrary evidence because he sees it only from his own point of view, and insists that there is always sufficient grounds for his ideas, even though they are insufficient and distorted. Therefore, we will try to pinpoint the limitations of this research result and deliver better results.

Reviewer 3 Report

The authors have satisfied my previous criticism. The authors should replace the word "increased" with the word "decreased" in line 299, based on the results in Fig. 3.

Author Response

Regarding the reviewer’s comments 3

The authors have satisfied my previous criticism. The authors should replace the word "increased" with the word "decreased" in line 299, based on the results in Fig. 3.

à We appreciate your comment. It's our mistake. According to the reviewer's comments, line 299 (changed line No. 265) was modified from "increase" to "decrease."